# EZH2 Methyltransferase Regulates Neuroinflammation and Neuropathic Pain

**DOI:** 10.3390/cells12071058

**Published:** 2023-03-31

**Authors:** Han-Rong Weng, Kyle Taing, Lawrence Chen, Angela Penney

**Affiliations:** Department of Basic Sciences, California Northstate University College of Medicine, Elk Grove, CA 95757, USA

**Keywords:** epigenetic, nociception, Schwann cells, analgesics, ncRNA

## Abstract

Recent studies by us and others have shown that enhancer of zeste homolog-2 (EZH2), a histone methyltransferase, in glial cells regulates the genesis of neuropathic pain by modulating the production of proinflammatory cytokines and chemokines. In this review, we summarize recent advances in this research area. EZH2 is a subunit of polycomb repressive complex 2 (PRC2), which primarily serves as a histone methyltransferase to catalyze methylation of histone 3 on lysine 27 (H3K27), ultimately resulting in transcriptional repression. Animals with neuropathic pain exhibit increased EZH2 activity and neuroinflammation of the injured nerve, spinal cord, and anterior cingulate cortex. Inhibition of EZH2 with DZNep or GSK-126 ameliorates neuroinflammation and neuropathic pain. EZH2 protein expression increases upon activation of Toll-like receptor 4 and calcitonin gene-related peptide receptors, downregulation of miR-124-3p and miR-378 microRNAs, or upregulation of Lncenc1 and MALAT1 long noncoding RNAs. Genes suppressed by EZH2 include suppressor of cytokine signaling 3 (SOCS3), nuclear factor (erythroid-derived 2)-like-2 factor (NrF2), miR-29b-3p, miR-146a-5p, and brain-specific angiogenesis inhibitor 1 (BAI1). Pro-inflammatory mediators facilitate neuronal activation along pain-signaling pathways by sensitizing nociceptors in the periphery, as well as enhancing excitatory synaptic activities and suppressing inhibitory synaptic activities in the CNS. These studies collectively reveal that EZH2 is implicated in signaling pathways known to be key players in the process of neuroinflammation and genesis of neuropathic pain. Therefore, targeting the EZH2 signaling pathway may open a new avenue to mitigate neuroinflammation and neuropathic pain.

## 1. Introduction

Neuropathic pain, caused by a lesion or disease of the somatosensory nervous system [1], is one of the most devastating clinical symptoms. Currently, management of neuropathic pain remains a challenge in clinics due to a lack of effective and safe analgesics. The development of novel analgesics with higher potency and safer features is an unmet need. Studies over the past three decades have shown that, under neuropathic conditions, pathologic glial cells play a critical role in aberrant neuronal activity in both the peripheral nerve and the spinal dorsal horn, termed peripheral and spinal central sensitizations, respectively [2]. Glial cells orchestrate neuronal activities in the peripheral and central nervous systems by releasing proinflammatory mediators (cytokines, chemokines, and others) [2,3,4]. In addition, glial cells in the CNS also control neuronal synaptic activities by clearing neurotransmitters via glial transporters [5,6]. Altered expression of signaling protein molecules is a critical mechanism underlying glial dysfunction and the genesis of neuropathic pain [7,8]. Identifying signaling molecules that regulate protein expression would provide potential targets for the development of analgesics.

Recent studies by us [9] and others [10,11,12,13,14,15] have shown that enhancer of zeste homolog-2 (EZH2), a key subunit of polycomb complex 2 (PRC2), plays an important role in the genesis of neuropathic pain by regulating gene expression. Protein expression is regulated by transcriptional and post transcriptional processes. Epigenetic regulation of gene expression is critically implicated in the genesis of chronic pain conditions [7,8]. It has been demonstrated that modification in DNA, RNA, or histones, as well as altered production of non-coding RNAs, is a crucial process that leads to aberrant gene expression in animals with neuropathic pain [7,8]. Post-translation modification in the N-terminal tail of histone—such as acetylation, methylation, phosphorylation, and ubiquitination—results in modification of gene expression. In general, histone acetylation at the lysine residue enhances gene transcription. However, histone methylation results in either the enhancement or the suppression of gene transcription, depending on which histone lysine residue is methylated [16,17]. EZH2 is a histone methyltransferase responsible for the methylation of histone 3 on lysine 27 amino acid position (H3K27), thereby suppressing gene transcription [18]. Accumulating studies have demonstrated that altered function of EZH2 and H3K27 methylation levels is critically implicated in the genesis of neuropathic pain by regulating neuroinflammation in the CNS and periphery. In this review, we summarize recent advances in this research area in the following sections: Section 2—EZH2 Biological function and its inhibitors; Section 3—Regulation of neuroinflammation and neuropathic pain by EZH2; Section 4—Upstream signaling molecules regulating EZH2 protein expression and function; Section 5—Downstream signaling molecules used by EZH2 to regulate neuroinflammation and neuropathic pain; Section 6—Mechanisms used by glia to cause aberrant neuronal activity along the pain signaling pathway in neuropathic pain; Section 7—Concluding remarks; and Section 8—Perspectives.

## 2. EZH2 Biological Function and Its Inhibitors

### 2.1. EZH2 Biological Function

Comprised of 746 amino acid residues [19], EZH2 is a catalytic subunit of polycomb repressive complex 2 (PRC2). EZH2 is present mainly in the nucleus but is also found in the cytoplasm [20]. PRC2 is a large multimeric complex composed of four core components: EZH2, embryonic ectoderm development (EED), retinoblastoma binding protein (RBBP4), and suppressor of zeste 12 homolog (SUZ12) [21]. Several other free proteins, including AEBP2 (adipocyte enhancer-binding protein 2), PCL (a polycomb-like protein), and JARID2 (jumonji and AT-rich interaction domain containing 2), are also found in PRC2 (Figure 1). EZH2 contains the Su(var)3-9, enhancer-of-zeste, and trithorax (SET) domain, which exerts the biological activity of methyltransferase via the C-terminal SET domain [22]. EZH2 catalyzes the addition of methyl groups to H3K27, with the assistance of the cofactor S-adenosyl-L-methionine (SAM), which provides the methyl group [21,23]. Enzymatic studies have shown that EZH2 facilitates the nucleophilic attack of the SAM cofactor by the lysine side chain in an S_N_2-like reaction, ultimately resulting in the transfer of the methyl group from SAM to the lysine residue [24] (Figure 2). Because the lysine side chain can be methylated up to three times on its terminal ammonium group, EZH2 is capable of mono-, di-, and tri-methylation of H3K27 [21,24]. However, EZH2 alone cannot function as a methyltransferase. Rather, its catalytic activity requires its interactions with other non-catalytic subunits, such as SUZ12 and EED [25,26]. While it is not required for methyltransferase activity, RBBP4 binds to the PRC2 complex through the N-terminal part of SUZ12 and leads to enhanced methyltransferase activity [26]. Although AEBP2, PCL, and JARID2 are not necessary for PRC2 enzymatic activity, these accessory proteins bind to the N terminal regions of SUZ12 and assist H3K27 methylation activity by PRC2 at specific sites in the genome [21,25].

PRC2 plays a critical role in epigenetic regulation due to its catalytic effects, causing methylation of H3K27. Methylation of H3K27 turns a lightly packed form of chromatin (euchromatin) into a condensed form of chromatin (heterochromatin) [27]. Heterochromatin limits the accessibility of the RNA polymerase II complex to the DNA, resulting in repression of gene transcription [28]. EZH2 is widely distributed in different tissues and cellular types of multiple systems, such as the nervous system [29], muscular system [30], endocrine system [31], and immune system [32], among others. As such, EZH2 plays a key role in many physiological and pathological processes, such as development [33], cell cycle [34], immune responses [35], inflammation [36,37], cardiovascular diseases [38], oncogenesis [27,39], and neurological disorders. EZH2 maintains normal neuronal function and survival by silencing genes responsible for Huntington’s disease [40,41] and Parkinson’s disease [41,42]. Deficient EZH2 activity has been implicated in the pathological processes of neurodegenerative diseases [40,41,42], while enhanced EZH2 activity contributes critically to neuroinflammation induced by subarachnoid hemorrhage [43] and neuropathic pain.

### 2.2. EZH2 Inhibitors

Development of EZH2 inhibitors has been driven by enhanced EZH2 activity being implicated in many diseases, including cancer [27,39]. EZH2 inhibitors can be divided into two groups: indirect inhibitors and direct inhibitors. EZH2 indirect inhibitors include 3-deazaneplanocin A (DZNep) and its less toxic analog form, named D9 [44]. During the formation of H3K27 methylation, the methyl group is catalytically transferred from a universal methyl donor, S-adenosyl-L-methionine (SAM), to the lysine side chains of the acceptor protein. The loss of a methyl group turns SAM into S-adenosyl-L-homocysteine (SAH), which is further metabolized by SAH hydrolase [23] (Figure 2). The EZH2 indirect inhibitors suppress the activity of SAH hydrolase, leading to an accumulation of SAH in cells, subsequently blocking the release of the methyl group from SAM and causing indirect inhibition of EZH2 methyltransferase activity [23,45]. As such, EZH2 indirect inhibitors are also called SAH hydrolase inhibitors. DZNep is the first identified EZH2 inhibitor that is not a selective EZH2 antagonist. Rather, DZNep is known to inhibit all methylation in histone and probably suppresses diverse methyltransferase [45,46].

The EZH2 direct inhibitors are a newer group of potent small molecules that produce their inhibitory effects by competing with the SAM binding site to the SET domain of EZH2 [18] (Figure 2). Thus, EZH2 direct inhibitors are also called SAM-competitive inhibitors. These inhibitors include EPZ005687, EI1, GSK126, UNC1999, GSK503, and EPZ-6438. One SAM-competitive inhibitor, Tazemetostat (Ipsen, Paris, France; Epizyme, Cambridge, MA, USA, was approved by the FDA in January 2020, making it the first EZH2 inhibitor drug for the treatment of lymphoma and sarcoma [47,48]. Many other SAM-competitive inhibitors have entered clinical trials for their anti-cancer effects, including GSK126 (GlaxoSmithKline, Middlesex, UK); CPI-1205 (MorphoSys, Boston, MA, USA); SHR2554 (HengRui Medicine Co., Ltd., Lianyungang, China); and PF-06821497 (Pfizer, New York, NY, USA) [17].

## 3. Regulation of Neuroinflammation and Neuropathic Pain by EZH2

### 3.1. Temporal Correlation between Enhanced EZH2 Protein Expression and Neuroinflammation Induced by Nerve Injury

The temporal correlation between changes in EZH2 protein expression and neuroinflammation induced by nerve injury was first revealed by us in 2017. We found increased global protein expression of EZH2 and trimethylation of H3K27 (H3K27me3) in the spinal dorsal horn in rats with neuropathic pain three and 10 days post-partial sciatic nerve ligation (pSNL) [9]. These levels were temporally accompanied by activation of microglia and astrocytes, as well as increased production of proinflammatory mediators, including tumor necrosis factor alpha (TNF-α), interleukin-1β (IL-1β), and monocyte chemoattractant protein-1 (MCP-1) in the same area [9]. Our findings were later confirmed by others who used different neuropathic pain models in mice and rats. In rats with neuropathic pain induced by chronic constriction injury (CCI), a significant enhancement in the global expression of both EZH2 and H3K27me3 in the spinal dorsal horn was found on days 1, 3, 5, 7, 10, and 14 post-operation in one study [10] and on days 3 to 21 post-nerve injury in another study [15]. Furthermore, the protein expressions of PRC2 subunits (EZH2, SUZ12 and EED) and TNF-α, IL-1β, and MCP-1 were upregulated in the spinal cord of mice 14 days after pSNL [13]. The EZH2 protein was expressed in neurons, microglia, and astrocytes in the spinal dorsal horn under normal conditions [9], whereas EZH2 protein expression in microglia was significantly increased three and 10 days after pSNL [9]. Furthermore, EZH2 protein and H3K27me3 were colocalized in microglia in the spinal dorsal horn and increased in rats five days after CCI [10].

EZH2 protein expression and alteration were also reported in the anterior cingulate cortex of the brain in rats with neuropathic pain induced by brachial plexus avulsion. It was found that EZH2 mRNA, EZH2 protein, and H3K27me3 protein expression was markedly elevated in the anterior cingulate cortex on days 3, 7, and 14 post-brachial plexus avulsion [11]. In immunohistochemical analysis, EZH2 overexpression was found to be co-localized with the Iba1 protein (a microglial marker) in the cingulate cortex, which was further confirmed in primary microglial cell culture. The increased EZH2 protein expression in the anterior cingulate cortex was temporally accompanied by microglial activation and over-production of TNF-α, IL-1β, and IL-6 in the same area [11]. More recently, in rats with neuropathic pain induced by CCI, it was found that activation of Schwann cells (the peripheral subset of glial cells) in the damaged sciatic nerve resulted in increased expression of EZH2 and H3K27me3, which was associated with upregulation of IL-1α, MCP-1, BDNF (brain derived neurotrophic factor) and GDNF (glial cell line-derived neurotrophic factor) in the same nerve on postoperative days 1 to 14 [12]. The temporal correlation between the increased EZH2 protein expression and neuroinflammation along the pain signaling pathway (the peripheral nerve, spinal cord, and anterior cingulate cortex) in animals with neuropathic pain highly suggests that EZH2 may regulate neuropathic pain by regulating neuroinflammation.

### 3.2. Inhibition of EZH2 Ameliorates Neuroinflammation and Neuropathic Pain

The role of EZH2 in the regulation of neuropathic pain and neuroinflammation in the CNS has been demonstrated in several neuropathic animal models by pharmacologically inhibiting EZH2 or by genetic engineering. It was reported that pre-existing mechanical allodynia and thermal hyperalgesia in rats induced by pSNL were attenuated by daily spinal inhibition of EZH2 with intrathecal administration of DZNep (20 nM in 10 mL) [9]. Furthermore, mechanical and thermal hyperalgesia induced by pSNL in rats was prevented when the animals received an intrathecal injection of either DZNep (20 nM in 10 mL) or GSK-126 (5 nM in 10 mL) one day prior to nerve injury and then daily for nine days [9]. Meanwhile, global protein levels of EZH2 and H3K27me3 in the spinal dorsal horn of neuropathic rats with pSNL were normalized after 10 days of intrathecal treatment with DZNep. In addition, activation of microglia and astrocytes and production of pro-inflammatory mediators (TNF-α, IL-1β, MCP-1) in the spinal cord induced by pSNL were all attenuated as well [9]. In rats with neuropathic pain induced by CCI, pre-emptive intrathecal [10] or intraperitoneal injection [12] of GSK126 prevented the development of mechanical and thermal hyperalgesia and reduced the enhanced protein expression of EZH2 and H3K27me3 in the spinal dorsal horn [10] and in the damaged sciatic nerve [12]. Intraperitoneal administration of GSK126 also reduced the levels of IL-1α, MCP-1, BDNF, and GDNF in the damaged sciatic nerve induced by CCI [12]. Furthermore, mechanical and cold allodynia induced by brachial plexus avulsion was attenuated when animals received systemic treatment with GSK-126 (7 mg/kg) (i.p.) daily from day 1 to 14 post-brachial plexus avulsion [11]. At the same time, the increased protein expression of EZH2, H3K27me3, TNF-α, IL-1β, and IL-6 in the anterior cingulate cortex induced by brachial plexus avulsion was reduced on days 3, 7, and 14 post-nerve injury.

The role of EZH2 in the regulation of neuroinflammation was further demonstrated in BV2 microglia and primary microglia from rats. It was shown that production of pro-inflammatory mediators (TNF-α, IL-1β, IL-6, IL-17, MCP-1) from BV2 microglia in response to lipopolysaccharide (LPS) stimulation was attenuated when EZH2 expression in BV2 microglia was knocked down with EZH2 siRNA [13,14]. Moreover, in BV2 microglial cells with EZH2 protein overexpression by plasmid vectors containing the EZH2 gene, microglial activation and production of pro-inflammatory mediators (TNF-α, IL-1β, MCP-1) from BV2 microglia induced by LPS stimulation were further exaggerated [13]. In rat primary microglial cells, LPS treatment increased expression of inducible nitric oxide synthase and CD16 and decreased expression of Arg-1 and CD206, indicating that activation of TLR4 promotes microglial M1-type polarization. These effects were reversed when the EZH2 gene was silenced, displaying M2-type polarization [49]. Collectively, these studies provided solid evidence to support EZH2 playing a crucial role in the regulation of neuropathic pain and neuroinflammation. The analgesic effects produced by systemic administration of the EZH2 inhibitors provide an easy route for the treatment of neuropathic pain in clinics.

## 4. Upstream Signaling Molecules Regulating EZH2 Protein Expression and Function

Studies in recent years have demonstrated that the protein expression and function of EZH2 in the CNS are regulated by signaling molecules that activate microglia [including Toll-like receptor 4 (TLR4) and calcitonin gene-related peptide (CGRP)] and non-coding RNAs.

### 4.1. TLR4

TLR4 is an innate immune pattern recognition receptor that is mainly expressed in microglia in the CNS, including the spinal cord [50,51]. TLR4 can be activated by endogenous molecules, including DNA-binding protein high-mobility group box 1 and cellular heat shock proteins, both of which are increased in the spinal cord after nerve injury [52,53,54]. TLR4 is also activated by LPS, an endotoxin from Gram-negative bacteria [55]. Microglial activation induced by TLR4 activation triggers a cascade of signaling pathways, leading to the overproduction of pro-inflammatory mediators. TLR4 activation in the spinal dorsal horn is implicated in the genesis of many pathological pain conditions, such as those induced by nerve injury [56,57,58], bone cancer [59,60], peripheral tissue inflammation induced by complete Freund’s adjuvant [61], and paclitaxel-induced neuropathic pain [62,63]. It was demonstrated that activation of TLR4 in BV2 microglia with LPS resulted in the activation of microglia, increased protein expression of PRC subunits (EZH2, SUZ12, and EED) in microglia, and the production of TNF-α, IL-1β, and MCP-1 [13]. LPS stimulation also increased EZH2 expression in primary microglia isolated from rats [49]. The production of pro-inflammatory cytokines in BV2 microglia and primary microglial cells was suppressed when EZH2 protein expression was suppressed by siRNA [13,49]. These findings provide evidence that EZH2 protein expression in microglia is positively regulated by the TLR4 signaling pathway.

### 4.2. CGRP

CGRP is another signaling molecule reportedly to be a positive modulator of EZH2 function. CGRP is a 37-amino acid neuropeptide released from nociceptive primary afferents. In addition to causing vasodilation and sensitizing nociceptors [64], CGRP in the spinal cord induces microglial activation by acting on several microglial CGRP receptors [calcitonin receptor-like receptor (CRLR), receptor activity-modifying protein 1 (RAMP1), and receptor component protein (CRCP)] [65,66]. It has been known that enhanced CGRP signaling in the spinal dorsal horn is a crucial mechanism contributing to the genesis of neuropathic pain [67,68,69]. Increased expression of CGRP and microglial activation were temporally associated with increased EZH2 and H3K27me3 expression in the spinal cord in neuropathic rats induced by CCI on days 1 to 10 after nerve injury [10]. Intrathecal administration of either CGRP antagonists (CGRP8-37, 2 μM) or EZH2 inhibitors (GSK-126, 5 nM) attenuated the development of mechanical allodynia and thermal hyperalgesia induced by CCI, along with suppression of EZH2 protein expression in the spinal dorsal horn [10]. Intrathecal administration of CGRP resulted in increased expression of both EZH2 and H3K27me3 in the spinal dorsal horn [10]. BV2 microglia expressed all CGRP receptor components: CRLR, RAMP1, and CRCP [10]. CGRP treatment enhanced protein expression of both EZH2 and H3K27me3 in BV2 microglia in a time-dependent manner with a maximal effect at four to six hours of treatment. The effects induced by CGRP on EZH2 and H3K27me3 were attenuated in the presence of either protein kinase A (PKA) or protein kinase C (PKC) antagonists [10]. Together, studies have indicated that CGRP enhances EZH2 protein expression via PKA and PKC.

### 4.3. miR-124-3p

Recent studies showed that microRNA-124-3p (miR-124-3p) in the spinal dorsal horn negatively regulates the genesis of neuroinflammation and neuropathic pain via EZH2 [70]. miR-124-3p belongs to the microRNA (miRNA) family, which is one type of non-coding RNA (ncRNA). miRNAs are 20-25 nucleotides in length [71]. Most miRNAs undergo several levels of processing before becoming mature miRNAs, when they interact with the 3′ or 5′ untranslated regions, coding sequences, or gene promoters to induce degradation, repress translation, or otherwise modulate gene expression [72]. It was reported that miR-124-3p levels in the spinal dorsal horn and spinal primary microglia were decreased in rats with neuropathic pain induced by CCI [70]. Upregulation of spinal miR-124-3p levels with a lentivirus vector alleviated both mechanical allodynia and thermal hyperalgesia in rats induced by CCI. Overexpression of miR-124-3p by a lentivirus miR-124-3p vector attenuated the production of TNF-α, IL-1β, and IL-6 in the spinal cord of rats induced by nerve injury and in primary microglial cells induced by LPS [70]. Through bioinformatic and luciferase analyses, it was demonstrated that miR-124-3p suppressed EZH2 protein expression by binding to the 3′-untranslated region (3′-UTR) of EZH2. Furthermore, the inhibitory effects of a lentivirus vector for miR-124-3p on neuropathic pain and production of TNF-α, IL-1β, and IL-6 were negated by a lentivirus vector for EZH2, which caused overexpression of EZH2 in the spinal cord. Together, these findings indicate that the loss of inhibitory effects by miR-124-3p on the UTR of EzH2 leads to increased protein expression of EZH2 and overproduction of pro-inflammatory cytokines and neuropathic pain.

### 4.4. miR-378

miR-378 has been reported to regulate EZH2 protein expression and neuropathic pain. It was demonstrated that miR-378 levels were decreased, while mRNA and protein expressions of EZH2 in the spinal dorsal horn were increased in rats with neuropathic pain induced by CCI on days 3, 7, 14, and 21 post-nerve injury [15]. Bioinformatic and luciferase analyses showed that miR-378 bound to the 3′-untranslated region (3′-UTR) of EZH2 in microglial N9 cells and 293 T cells. Overexpression of miR-378 induced by intrathecal injection of a lentivirus miR-378 vector attenuated increased EZH2 protein expression, as well as mechanical and thermal hyperalgesia induced by CCI. These data indicate that spinal protein expression of EZH2 and neuropathic pain are negatively regulated by miR-378 [15].

### 4.5. Lncenc1

It has been demonstrated that embryonic stem cell expressed 1 (Lncenc1) regulates EZH2 protein expression and neuropathic pain [13]. Lncenc1 is a long ncRNA (LncRNA), which is defined as RNA molecules longer than 200 nucleotides and not translated into functional proteins [73]. It has been shown that, depending on their localization and specific binding interactions with DNA, RNA, and proteins, lncRNAs can modulate gene expression and interfere with signaling pathway activities [73]. Lncenc1 is highly abundant in naïve embryonic stem cells and plays a role in self-renewal [74]. It was reported that Lncenc1 expression was increased in the dorsal root ganglion (DRG) of rats with neuropathic pain induced by pSNL 14 days after nerve injury [13]. At the same time, microglial activation and increased protein expressions of TNF-α, IL-1β, MCP-1, and PRC2 subunits (EZH2, SUZ12, and EED) were found in the spinal dorsal horn. In BV2 microglia, EZH2 was found to be an RNA protein binding to Lncenc1 [13]. Overexpression of Lncenc1 in BV2 microglia cells resulted in increased protein expressions of PRC2 components (EZH2, SUZ12, and EED) and production of TNF-α, IL-1β, and MCP-1. In contrast, knockdown of Lncenc1 produced the opposite effects. Further experiments demonstrated that knockdown of Lncenc1 in the spinal cord with Lncenc1 siRNA attenuated mechanical allodynia and thermal hyperalgesia induced by pSNL and the production of TNF-α, IL-1β, and MCP-1 in the spinal dorsal horn induced by nerve injury [13]. It is unknown whether Lncenc1 expression in the spinal cord is altered by nerve injury given that this study only observed the changes of Lncenc1 in the DRG. Mechanisms about how Lncenc1 increases EZH2 protein expression remain open for further exploration.

### 4.6. MALAT1

Metastasis-associated lung adenocarcinoma transcript 1 (MALAT1) is a long ncRNA, also known as nuclear-enriched abundant transcript 2 (NEAT2). MALAT1 recently has been shown to be involved in the genesis of neuropathic pain and enhance the function of EZH2. Rats with neuropathic pain induced by CCI had increased expression of MALAT1 in the spinal dorsal horn [75,76]. Mechanical allodynia and thermal hyperalgesia in neuropathic rats were attenuated when MALAT1 in the spinal cord was suppressed by MALAT2 knockdown or MALAT1 antagonists [75,76]. At the same time, suppression of MALAT1 resulted in reduced production of cyclooxygenase-2, IL-1β, and IL-6 in the spinal cord [75,76]. Interestingly, another study reported that expression of MALAT1 was increased in BV2 microglia treated with LPS and ATP [77]. Knockdown of MALAT1 attenuated inflammatory responses induced by LPS/ATP treatment in BV2 cells, along with suppression of H3K27me3 but not EZH2. Further experiments demonstrated that MALAT1 and EZH2 were bound together. MALAT1 knockdown significantly reduced the binding of EZH2 to its target gene, nuclear factor (erythroid-derived 2)-like-2 factor (Nrf2) [77]. Hence, it was suggested that increased expression of MALAT1 in microglia induced by LPS/ATP enhances the function of EZH2 by promoting EZH2 binding to its target gene to regulate inflammatory responses.

## 5. Downstream Signaling Molecules Used by EZH2 to Regulate Neuroinflammation and Neuropathic Pain

Methylation of H3K27 results in compression of chromatin and suppression of gene transcription [78]. As mentioned previously, increased expression of EZH2 and H3K27me3 in animals with neuropathic pain is correlated with microglial activation and overproduction of proinflammatory mediators. Thus, the genes repressed by increased activity of EZH2/H3K27me3 must be anti-inflammatory mediators that negatively regulate neuroinflammation. In the context of neuroinflammation and neuropathic pain, several genes have been reported to be regulated by EZH2. These genes include miR-29b-3p [49], miR-146a-5p, suppressor of cytokine signaling 3 (Socs3) [79], nuclear factor (erythroid-derived 2)-like-2 factor (Nrf2) [77], and brain-specific angiogenesis inhibitor 1 (BAI1).

### 5.1. miR-146a-5p/HIF-1α

Recent studies showed that EZH2 in microglia regulates neuroinflammation and neuropathic pain induced by spinal cord injury via miR-146a-5p. Downregulation of miR-146a-5p in the spinal cord has been reported in rats with neuropathic pain induced by spinal cord injury or peripheral nerve injury, while enhancing miR-146a-5p expression attenuated neuropathic pain by suppressing TNF receptor associated factor 6 (TRAF6) signaling [14,80]. Spinal cord injury resulted in increased protein expression of EZH2 and H3K27me3 in the spinal cord [14]. Similarly, BV2 microglia treated with LPS showed increased protein expression of EZH2 and H3K27me3 with downregulation of miR-146a-5p [14]. Enrichment of H3K27me3 was found in the miR-146a-5p promoter region by ChIP analysis. miR-146a-5p expression was increased, while production of TNF-*α*, IL-6, and IL-17 was reduced in rats with spinal cord injury treated with EZH2 inhibitors [14]. Further bioinformatic analysis and dual-luciferase report assay showed that miR-146a-5p bound to the mRNA of hypoxia-inducible factor 1*α* (HIF-1*α*) [14]. Transfection of miR-145-5p mimics into microglia reduced HIF-1*α* expression [14]. Inhibition of EZH2 also suppressed the enhanced expression of HIF-1*α* induced by spinal injury [14]. HIF-1*α* is a transcription factor known to reprogram cellular metabolism and to be a positive regulator of inflammation [14,81,82]. Collectively, this study demonstrated that EZH2 induces neuroinflammation by suppressing miR-146a-5p gene expression, resulting in increased HIF-1*α* expression and neuroinflammatory responses.

### 5.2. miR-29b-3p/MMP-2

It was found that enhanced EZH2 and H3K27me3 expression in rat primary microglia treated with LPS was associated with reduced expression of miR-29b-3p, along with increased production of pro-inflammatory cytokines [49]. ChIP assay demonstrated that EZH2 bound to the promoter region of miR-29b-3p, and silencing of EZH2 reduced the enrichment of EZH2 and H3K27m3 on the miR-29b-3p promoter and production of pro-inflammatory cytokines in microglia induced by LPS. Further bioinformatic analysis predicted the binding of miR-29b-3p to matrix metalloproteinase-2 (MMP-2) mRNA; this prediction was then confirmed by dual-luciferase reporter gene assay. MMP2 mRNA levels were increased in microglial cells treated with LPS but reduced upon silencing of EZH2. Inhibition of miR-29b-3p enhanced MMP2 transcription level and pro-inflammatory cytokine levels. Taken together, these studies demonstrated that EZH2/H3K27me3 suppressed miR-29b-3p expression by binding to the miR-29b-3p promoter, which subsequently enhanced MMP2 transcription and production of pro-inflammatory cytokines from microglia. It is noteworthy that upregulation of MMP-2 in the spinal dorsal horn was found in animals with nerve injury, while treatment with MMP-2 inhibitor attenuated allodynia [83,84].

### 5.3. SOCS3

The suppressor of cytokine signaling 3 protein (SOCS3) is a negative regulator of the Janus kinase (JAK)/signal transducer and activator of transcription 3 (STAT3) pathway. Activation of the JAK/STAT3 signaling pathway and suppression of SOCS 3 expression were reportedly implicated in the development of neuropathic pain induced by CCI [85,86]. Inhibition of the JAK/STAT3 signaling pathway prevented the abnormal expression of IL-6 and MCP-1 in the spinal cord induced by CCI, along with suppression of mechanical allodynia in rats [86]. In cultured macrophage cells, it was shown that EZH2 enriched the transcription start site (TSS) and distal enhancer regions of the SOCS3 mRNA [79]. This finding was further confirmed by EZH2 depletion causing a substantial reduction in H3K27me3 levels at both the TSS proximal region and the distal enhancer of the SOCS3 mRNA. Deletion of the EZH2 gene resulted in significant increases in both mRNA and protein levels of SOCS3 in macrophages and BV2 microglia [79]. The beneficial effects induced by EZH2 inhibitors on neuroinflammation in the rat brain and macrophages treated with LPS were attenuated when gene expression of the SOCS3 was knocked down by siRNA [79]. It was further demonstrated that the suppression of SOCS3 gene expression by EZH2/H3H27me3 leads to suppression of Lys48-linked ubiquitination and degradation of TRAF6, resulting in enhancement of TLR4-induced MyD88-dependent NF-κB activation and inflammatory gene expression from microglia [79].

### 5.4. Nrf2

Nuclear factor erythroid derived-2-related factor 2 (Nrf2) is a transcription factor that positively regulates antioxidant and detoxification genes [87]. Activation of the Nrf2 pathway produces analgesic effects in animals with neuropathic pain [87]. It was demonstrated in BV2 microglia that EZH2 bound to the Nrf2 promoter, resulting in inhibition of Nrf2 expression [77]. The binding of EZH2 to the Nrf2 gene was increased when BV2 microglial cells were stimulated by LPS/ATP or by MALAT1, a long ncRNA that induces microglial and inflammasome activation and reactive oxygen species (ROS) production. Further experiments showed that recruitment of EZH2 to the promoter of Nrf2 resulted in the suppression of Nrf2 expression and neuroinflammation [77]. In summary, the enhanced expression of EZH2 induced by microglial activation leads to suppression of Nrf2 gene expression, leading to enhanced inflammasome activation and ROS.

### 5.5. BAI1

Brain-specific angiogenesis inhibitor 1 (BAI1) is a transmembrane adhesion G-protein coupled receptor that recognizes phosphatidylserine. Phosphatidylserine is a plasma membrane lipid that is ‘flipped out’ and exposed on the surface of apoptotic cells [88,89]. BAI1 binds to phosphatidylserine and promotes autophagy by microglia [90]. Autophagy is a physiological process involving the decomposition of intracellular molecules and organelles and self-digestion. Autophagy is essential for cellular differentiation, homeostasis, and survival [91,92] and is conserved across evolution. Recent studies have shown that altered autophagy is involved in neuroinflammation processes in many neurological diseases, including Parkinson’s disease [93,94], Alzheimer’s disease [95,96], and neuropathic pain [67,97,98]. It was reported that autophagic activity in microglia in the spinal cord was suppressed in animals with neuropathic pain from day 2 to day 28 after nerve injury [99]. Enhancement of autophagic activity could suppress pain behavior by suppressing inflammatory responses [99]. It was demonstrated that BAI1 protein expression in the spinal dorsal horn was downregulated in mice with chronic pain induced by pSNL and in BV2 microglia treated with LPS [13]. Using ChIP techniques, it was shown that both EZH2 and H3K27me3 were enriched in the promoter region of the BAI1 gene. Overexpression of BAI1 prevented the production of TNF-α, IL-1β, and MCP-1 in BV2 microglia induced by LPS or by overexpression of Lncenc1. Similarly, knockdown of Lncenc1 in the spinal cord prevented the downregulation of BAI1 and the production of pro-inflammatory cytokines, as well as mechanical allodynia and thermal hyperalgesia induced by pSNL in mice [13].

Regulation of autophagy by EZH2 was demonstrated in another study. It was shown that microglial autophagy activity in the anterior cingulate cortex was suppressed in rats with neuropathic pain caused by brachial plexus avulsion on days 3, 7, and 14 after surgery [11]. This finding was concomitantly associated with increased EZH2 and H3K27me3 in the same area. Pharmacological inhibition of EZH2 with GSK126 treatment (i.p.) decreased TNF-α, IL-1β, and IL-6 protein expression, as well as increasing autophagy function, along with a reduction in mechanical allodynia and cold allodynia. These effects were blocked when autophagy function was pharmacologically suppressed, indicating that EZH2 aggravates neuroinflammation and neuropathic pain via suppression of autophagy. Further experiments demonstrated that EZH2 suppressed autophagic activity by activating the MTOR (mammalian target of rapamycin)-dependent signaling pathway [11].

## 6. Mechanisms Used by Glia to Cause Aberrant Neuronal Activity along the Pain Signaling Pathway in Neuropathic Pain

EZH2 is implicated in the genesis of neuropathic pain mainly via regulation of glial activation and production of pro-inflammatory mediators in the peripheral and central nervous systems. Mechanisms used by glial cells to cause central sensitization in the spinal cord have been reported by us and others. Activation of excitatory glutamatergic synapses depends on three key determinants: the amount of glutamate released from presynaptic terminals, the function of postsynaptic glutamate receptors, and the rate of glutamate re-uptake by glutamate transporters [100]. These three determinants are all regulated by glial cells. For example, peripheral nerve injury leads to the activation of glial cells and subsequent release of pro-inflammatory mediators (such as IL-1β, TNF-α, and MCP-1) in the spinal dorsal horn [4,9,11,101]. At the pre-synaptic level, IL-1β enhances glutamatergic synaptic activities at the first synapses between the primary nociceptive afferents and dorsal horn neurons in the spinal dorsal horn by promoting glutamate release from presynaptic terminals^2^. IL-1β facilitates presynaptic glutamate release by enhancing presynaptic NMDA receptor activity [2,102]. At the postsynaptic level, IL-1β augments post-synaptic AMPA and NMDA receptor activities [4,62,103]. Likewise, glutamatergic synaptic activities are also increased by TNF-α [104] and MCP-1 [105] in a similar fashion. Glutamate released from presynaptic terminals and the homeostasis of extracellular glutamate are mainly maintained by glial glutamate transporters located on the plasma membrane of astrocytes [100]. Deficiency in glial glutamate transporters has been reported in animals with neuropathic pain [6,106,107,108]. We have shown that IL-1β reduces astrocytic glial glutamate transporter function by activating protein kinase C and promoting glutamate transporter endocytosis from the cell surface into the cytosol [108]. Dysfunction of glial glutamate transporters prolongs the activation time of AMPA and NMDA receptors to peripheral nociceptive sensory stimulation [5,6,107]. Severe deficiency of glial glutamate transporters can even cause glutamate release from astrocytes and subsequently trigger glutamatergic interactions between neurons and astrocytes [109]. Furthermore, suppression of glial glutamate transporters by IL-1β reduces presynaptic GABA (a key inhibitory neurotransmitter) synthesis via the glutamate-glutamine cycle [110]. Postsynaptic GABAergic and glycinergic receptor activities are also reduced by IL-1β, IL-6 and TNF-α [103] as well. Collectively, pro-inflammatory mediators in the spinal dorsal horn enhance neuronal activities by augmenting excitatory synaptic activities and suppressing inhibitory synaptic activities, ultimately contributing to the central mechanisms underlying the genesis of neuropathic pain.

Overproduction of proinflammatory mediators (such as BDNF, MCP-1, GDNF, IL-1α) in the peripheral nerve causes aberrant neuronal activity in the peripheral nerve (peripheral sensitization). For example, BDNF released from Schwann cells can act on trkB receptors in nociceptors, leading to increased activity of PKC [111]. Increased PKC in nociceptive sensory neurons is known to cause sensitization of nociceptors by phosphorylating TRPV1 receptors [112,113]. TRPV1 receptors are also sensitized by MCP-1 [114]. GDNF can induce heat hyperalgesia and robustly sensitized cold responses in a TRPM8-dependent manner in mice [115], while IL-1α promotes the production and release of substance P in sensory neurons [116]. Substance P released at the periphery induces plasma extravasation (neurogenic inflammation) and peripheral sensitization [117,118]. In short, by regulating the production of pro-inflammatory mediators, EZH2 regulates peripheral and central sensitization induced by nerve injury.

## 7. Concluding Remarks

In this review, we summarized recent advances for the role of EZH2 in the regulation of neuroinflammation and neuropathic pain. In animals with neuropathic pain, EZH2 protein expression was enhanced in the spinal dorsal horn, anterior cingulate cortex, and the damaged nerve. Inhibition of EZH2 with DZNep or GSK-126 produced analgesic effects in animals with neuropathic pain. While EZH2 is expressed in neurons, astrocytes, and microglia in the CNS and Schwann cells in the peripheral nerve, its role in the regulation of neuropathic pain has been best studied in the context of microglia and the production of pro-inflammatory cytokines and chemokines. EZH2 and H3K27me3 expression in microglia was significantly increased in animals with neuropathic pain induced by nerve injury. EZH2 mediated production of pro-inflammatory cytokines in microglia induced by activation of TLR4 or CGRP. EZH2 was also engaged the pro-inflammatory effects caused by altered expression of several ncRNAs following nerve injury. EZH2 produced the pro-inflammatory effects via methylation of H3K27 and subsequently suppressed the anti-inflammatory genes that code protein molecules and microRNA. The genes suppressed by EZH2/H3K27me3 belong to anti-inflammatory mediators and are known to be negative regulators of the genesis of neuropathic pain. Through suppressing the expression of anti-inflammatory genes, the enhanced EZH2 function facilitates microglial activation and production of pro-inflammatory cytokines and chemokines, ultimately causing excessive neuronal activation along the pain signaling pathway and leading to the development of neuropathic pain. Figure 3 summarizes our current understanding of the regulation of neuroinflammation and neuropathic pain by EZH2. The upstream signaling molecules regulating EZH2 protein expression and function and downstream signaling molecules used by EZH2 in microglia to regulate neuroinflammation and neuropathic pain are shown. Given that EZH2 is engaged in multiple signaling pathways known to be key players in the process of neuroinflammation and genesis of neuropathic pain, inhibition of the EZH2 signaling pathway may offer a promising new treatment for neuroinflammation and neuropathic pain. In light of overproduction of pro-inflammatory cytokines being implicated in the genesis of neuropathic pain induced by nerve injury [56,57,58] or chemotherapy [62,63], pain induced by bone cancer [59,60], and peripheral tissue inflammation [61], it is conceivable that inhibition of EZH2 may produce analgesic effects on pathological pain conditions with different etiologies.

## 8. Perspectives

Our current understanding of the signaling molecules regulating EZH2 function and expression and the signaling molecules used by EZH2 to alter neuronal activity along the pain-signaling pathway is still at the early stage. Many questions remain to be answered. For example, changes in EZH2 expression induced by nerve injury were also found in spinal dorsal horn neurons [9] and in sensory neurons in the DRG [119], in addition to those in microglia and Schwann cells. The functional implications of such changes for pain signaling transmission remain elusive. While production of proinflammatory mediators in Schwann cells in the damaged nerve is known to be controlled by EZH2, genes repressed by EZH2 in Schwann cells have still not been identified. Furthermore, it has been demonstrated that other subunits in PRC2 play a regulatory and supplementary role to EZH2 [25,26]. Given that other subunits in PRC2 undergo plastic changes in their protein expression under neuropathic pain or upon microglial activation [13], identifying signaling pathways that regulate the protein expression of other subunits in the PRC2 may provide novel molecular targets for controlling neuroinflammation and neuropathic pain. Finally, current studies of the role of EZH2 in the regulation of neuropathic pain and neuroinflammation focus mainly on its canonical role in H3K27 methylation in the nucleus. It is known that EZH2 also exerts other biological effects via its non-canonical action. For example, EZH2 in the cytoplasm can cause methylation of cytoplasmic proteins (such as Vav1 [20] protein, cytosolic talin [120]) and alter cell migration, invasion, proliferation, and adhesion. EZH2 is involved in DNA methylation via recruiting DNA methyltransferases (DNMTs) [121]. EZH2 can also exert its biological action independently of PRC2. For instance, phosphorylation of EZH2 at S21 mediated by the PI3K/AKT pathway can switch its function from a transcriptional repressor to become a transcriptional co-activator [122]. Further investigation into the EZH2 signaling pathway in the regulation of neuropathic pain would yield novel molecular targets for the development of analgesics and provide a basis for repurposing EZH2 inhibitors for the management of neuropathic pain, given that EZH2 inhibitors have been approved by the FDA [47] and undergone clinical trials [17] for cancer treatment.

## Figures and Tables

**Figure 1 cells-12-01058-f001:**
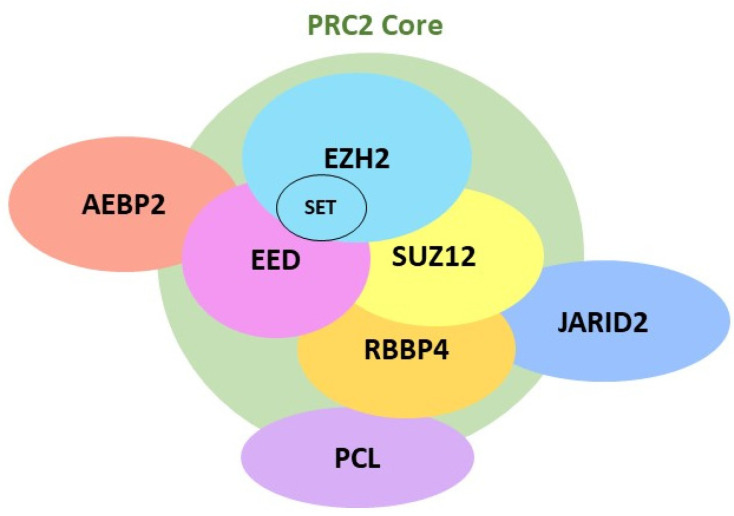
Assembly of PRC2 core and non-core subunits. EZH2 contains the SET domain, which exerts methyltransferase activity on the PRC2 complex. EED: embryonic ectoderm development; EZH2: enhancer of zeste homolog 2; RBBP4: retinoblastoma binding protein 4, SUZ12: suppressor of zeste 12 homolog; SET: Su(var)3-9, enhancer-of-zeste, and trithorax domain; AEBP2: adipocyte enhancer-binding protein 2; PCL: polycomb-like proteins; JARID2: jimonji, AT-rich interactive domain 2.

**Figure 2 cells-12-01058-f002:**
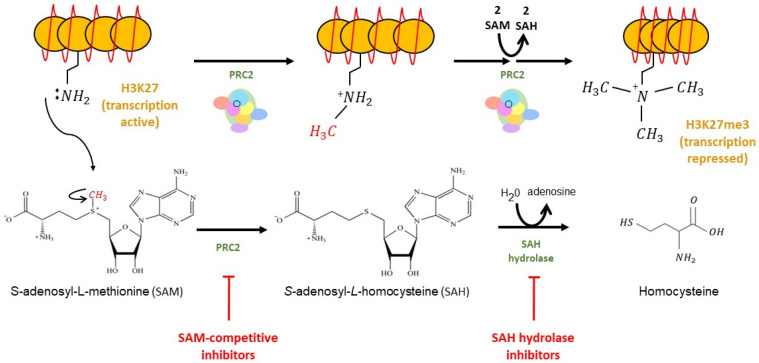
Model of H3K27 methylation by PRC2. PRC2 catalyzes the nucleophilic attack of the SAM cofactor, resulting in the transfer of the methyl group (in red) from SAM to the lysine residue of H3K27. This transfer repeats two more times to produce a tri-methylated lysine-27 residue of histone 3 (H3K27me3), which ultimately results in a more tightly packed heterochromatin complex that represses gene transcription. SAH is further metabolized into adenosine and homocysteine via SAH hydrolase. The catalytic subunit (EZH2) of PRC2 and SAH hydrolase can be inhibited by SAM-competitive inhibitors and SAH hydrolase inhibitors, respectively. SAM: *S*-adenosyl-L-methionine; SAH: *S*-adenosyl-L-homocysteine.

**Figure 3 cells-12-01058-f003:**
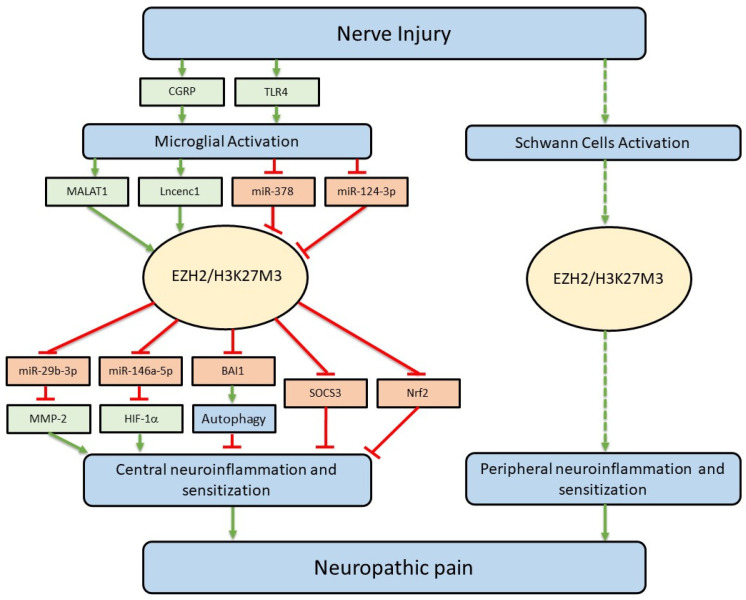
The EZH2 signaling pathway in the regulation of neuroinflammation and neuropathic pain. Upstream signaling molecules regulating EZH2 protein expression and function and downstream signaling molecules used by EZH2 in microglia (left) are shown. The EZH2 signaling pathways in Schwann cells (right) remain unknown (shown in dash line). Genes with expression or activity that is enhanced are indicated in green, while genes with expression or activity that is suppressed are in red.

## Data Availability

Not applicable.

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
