# Peer review of "EZH2 Methyltransferase Regulates Neuroinflammation and Neuropathic Pain"

_cells, 2023, doi:10.3390/cells12071058_

Round 1

Reviewer 1 Report

The main question addressed is the role of  EZH2 expression in glial cells  in neuropathic pain. 

The review is well written and summarizes results with regards to the role of EZH2 expression in glial cells in neuropathic pain. Increase in expression is suggested to increase inflammatory cytokines and pain. The review is divided in six parts, and mechanisms that regulate EZH2 expression as well as its function are well discussed. 1.  EZH2 Biological  function and its inhibitors;  2. Regulation of neuroinflammation and neuropathic pain by 63 EZH2;  3. Upstream signaling molecules regulating EZH2 protein expression and function;   4. Downstream signaling molecules used by EZH2 to regulate neuroinflammation and  neuropathic pain;  5. Mechanisms used by glia to cause aberrant neuronal activity along  the pain signaling pathway in neuropathic pain; 

6. Concluding remarks and prospectives.  

Since there are several articles on this topic this review is a good summary of the area and helps give a more clear view of the role of EZH2 in neuropathic pain.

One thing that should be mentioned in the concluding remarks  is that since EZH2 has many effects on gene regulation in several tissues, a better drug target than EZH2 could be one of EZH2 targets,  whose expression is decreased by EZH2 regulation or modulating specific expression of EZH2 in glial cells to have a more specific effect in neuropathic pain. 

The references are appropriate.

Figure 3 is a nice synthesis of the article.The effect of EZH2 on neurotransmittors could be included. One thing that should be mentioned is that since EZH2 has many effects on gene regulation in several tissues a better drug target that EZH2 could be one of EZH2 targets that are specific in neuropathic pain.

Reviewer 2 Report

Dear Author,

Thanks for submitting your research manuscript entitled “EZH2 methyltransferase as a therapeutic target for the treatment of neuroinflammation and neuropathic pain".

Note: Before giving my final comments, as well as the final revision of this manuscript,

Firstly, the author needs to address the following comments scientifically.

Major concerns:-Please find out the following comments

Title, Abstract, and Introduction:

-          Most of the data is copy paste from other resources. Plag percentage is very-very high. What is this???????????? Almost each paragraph and sections are fully loaded with highest similarity index.

-          The title needs to be very specific and is not acceptable in its current form.

-          Lack of update as well old & outdated references with incomplete review design is another major concern. 

-          Outdated References quoted in text with important perspective of neuropathic pain. Reviewer feels, its complete ignorance by authors.

-          The rationale and purpose behind the correlation between EZH2 methyltransferase as a therapeutic target for the treatment of neuroinflammation and neuropathic pain is not clear and incomplete throughout the manuscript.

-          Abstract is very poorly written.

-          Authors must start their introduction direct with the correlation and rationale behind this study instead of writing about the common information regarding pathological alterations and its etiopathological factors during the progression of neuropathic pain and related neurological impairments.

-          The abstract is very confusing. Irrational and fused with repetitions. Scientific output is not clear with this abstract.

Example 1: Glial activation and over-production of proinflammatory mediators are crucial mechanisms in the genesis of neuropathic pain. Recent studies by us and others have shown that enhancer of zeste homolog-2 (EZH2) (a histone methyltransferase) in glial cells regulates the genesis of neuropathic pain via modulating production of proinflammatory mediators.??????? Irrational lines without scientific output is another major concern.

-          The key messages and conclusion are very poorly explained. Reviewer surprise to see the justification in abstract conclusion “By analyzing the KKI network, we found five kinases as the topologically most important hub kinases that may serve as potential therapeutic targets in neuropathic pain as well as Given that EZH2 inhibitors have been approved by the FDA or in clinical trials for cancer treatment, further investigation into molecular mechanisms underlying the role of EZH2 in the genesis of neuropathic pain would provide a base to repurpose EZH2 inhibitors for the management of neuropathic pain.”. What is this???????? Author need to directly strike in scientific and readily manner.

-          Provide the separate future perspective for this manuscript.

-          The reviewer feels the author needs to elaborate and justify it with proper citations and strong evidence. The author fails to explain the relevant justification in the introduction as well as mentioned in the discussion part.

-          A major drawback is a lack of clinical evidence, and the preliminary experimental data of neuropathic pain and related neurological impairments with respect to selective enzymes.

-           The reviewer found irrational and non-scientific justification in the abstract—introduction and the discussion part.

-          Long paragraphs must be split into subheading according to content.

-          Authors fail to justify the correlation of review theme in conclusion regarding utilization of multiple therapies and methyltransferase interventions in the prevention and diagnosis of neuropathic pain.  

-          Author must prepare two or more figures showing the involvement of rationale and theme focus on this review especially cellular and molecular signaling kinases associated with progression of pain.

-          Without any significant molecular pathways and absence of figure make it difficult to further proceed.

-          Need to add a separate table focuses on clinical and pre-clinical relevant research papers focused on methyltransferase as a therapeutic target in neuroinflammation and neuropathic pain

Results 

-          There is lack of information regarding EZH2 inhibitors employed in neuropathic pain. Need to add more data in table form also.

-          Results need more clarification and significant justification in review sub-sections paragraphs. Differentiating between the outcome and the discussion sections is quite difficult.

-          Need major revision. Need good quality figure with clear objectives?

Discussion:

-     To address the outcome of measures/results separately and how they correlate with the existing literature, it would be better if the author restructured to take a more critical approach to prevent neuropathic pain and associated with neuronal impairments and targeted with current therapies targeting EZH2 methyltransferase.
-     In both the discussion and the conclusion, the aims, rationale, and future perspectives are not evident clearly in relation to previously published in-vitro and in-vivo experimentation.
-     The discussion is usually organized at the beginning to address all the observations and evaluate them at the end. It makes the results easier to contextualize and simpler to comprehend.

- Furthermore, a minimal critical analysis should be provided regarding EZH2 methyltransferase associated neurological complications.

- Add limitations of this review at the end of the discussion part.

Conclusion:

-          Need to revise the conclusion in a scientific manner. Not accepted in its current form.

-          This reviewer considers that this paper cannot be published in its present form. A detailed revision shortening, ordering and following the commented ideas could improve this interesting paper in a meaningful manner.

-          Several typewriting mistakes are present and need correction. This reviewer remains at entire disposal for the next version.

Round 2

Reviewer 2 Report

Dear Author, 

After careful revision, revised manuscript can proceed further for publication.